# Genetic testing in individuals with extreme HDL-C levels: Diagnostic yield and clinical implications from the Tromsø Study

Åsa Schawlann Ølnes [1,2‡]*, Marianne Teigen [1,2‡], Thea Bismo Strøm [1], Erik Kristoffer Arnesen [3], Antoine Rimbert [4], Anne Elise Eggen [5], Katrine Bjune [1]

1 Unit for Cardiac and Cardiovascular Genetics, Department of Medical Genetics, Oslo University Hospital, Oslo, Norway, 2 Faculty of Medicine, University of Oslo, Oslo, Norway, 3 Department of Nutrition, Institute of Basic Medical Sciences, University of Oslo, Oslo, Norway, 4 Nantes Université, CNRS, INSERM, l'institut du thorax, Nantes, France, 5 Department of Community Medicine, UiT, The Arctic University of Norway, Tromsø, Norway

‡ These authors contributed equally and share first authorship.
* aasoel@ous-hf.no

## Abstract

### Aim

This study aimed to assess the prevalence of genetic variants responsible for extreme levels of high-density lipoprotein cholesterol (HDL-C) and evaluate the adequacy of current thresholds for genetic testing of HDL-related dyslipidemia.

### Methods

Using data from the Tromsø Study, a population-based cohort in Northern Norway, we identified 210 individuals with HDL-C levels ≤ 0.5 mmol/L or ≥ 3.0 mmol/L. Six HDL-related genes (*ABCA1, APOA1, CETP, LCAT, PLTP, SCARB1*) were sequenced in these participants. We classified variants according to ACMG guidelines, incorporating functional assays and UK Biobank data for additional phenotype-genotype associations.

### Results

We identified 38 variants of interest across six HDL-related genes, of which 10 were considered potentially causative, found in 14 individuals. Genetic causes were detected in 33.3% of individuals with low HDL-C and 5.05% of those with high HDL-C. Sex-specific analyses showed that using HDL-C thresholds aligned with population distributions improved detection of individuals with pathogenic variants, particularly among women with high HDL-C and men with low HDL-C. These findings suggest that current uniform thresholds may overlook clinically relevant cases and that incorporating sex-specific HDL-C distributions could enhance the identification of individuals with suspected genetic HDL disorders.

**Citation:** Ølnes ÅS, Teigen M, Strøm TB, Arnesen EK, Rimbert A, Eggen AE, et al. (2026) Genetic testing in individuals with extreme HDL-C levels: Diagnostic yield and clinical implications from the Tromsø Study. PLoS One 21(4): e0344627. https://doi.org/10.1371/journal.pone.0344627

**Data availability statement:** All relevant data are within the paper and its Supporting information file.

**Funding:** This research was supported by Nasjonalforeningen for Folkehelsen (Norwegian Health Association) (https://nasjonalforeningen.no/) (Grant No. 22741, received by K.B.). The funder had no role in study design, data collection and analysis, decision to publish, or preparation of the manuscript.

**Competing interests:** The authors have declared that no competing interests exist.

## Conclusions

Genetic testing for HDL-related dyslipidemia is underutilized, with many individuals not meeting the current extreme HDL-C threshold criteria. Revised sex-specific thresholds for genetic testing will improve the identification of pathogenic variants and provide more accurate diagnoses of HDL-related disorders. Continued research is essential to refine our understanding of HDL genetics and its clinical implications.

---

## Introduction

The strong observational inverse association between high-density lipoprotein cholesterol (HDL-C) levels and cardiovascular disease (CVD) risk has, over the last two decades, encouraged several therapeutic efforts to increase HDL-C levels [1,2]. Whereas previous trials have failed to deliver the expected protective effects against CVD, the focus of the new generation components has shifted from increasing the atheroprotective HDL-C levels to decreasing the atherogenic apolipoprotein B-containing lipoproteins [3–7]. Furthermore, while HDL-C is often referred to as "good cholesterol" in clinical contexts, recent population studies have raised concerns that exceptionally high HDL-C levels may paradoxically be linked to increased mortality, underscoring the complexity of HDL dynamics [8–11].

HDL plays a crucial role in reverse cholesterol transport, the transport of cholesterol from peripheral tissues to the liver for excretion. The biogenesis of HDL begins with the ATP-binding cassette transporter A1 (ABCA1), which effluxes cellular phospholipids and cholesterol to apolipoprotein A1 (ApoA1), forming nascent discoidal pre-β-HDL [12]. This precursor is subsequently converted into mature spherical α-HDL through the action of lecithin-cholesterol acyltransferase (LCAT), which esterifies free cholesterol within the discoidal HDL, allowing it to be sequestered in the core and creating a more spherical HDL particle [13].

As HDL matures and accumulates cholesterol, it becomes a substrate for both cholesteryl ester transfer protein (CETP) and phospholipid transfer protein (PLTP). CETP mediates the exchange of cholesteryl esters and triglycerides between HDL and low-density lipoprotein (LDL) or very-low-density lipoprotein (VLDL), transferring cholesteryl esters from HDL to these particles and triglycerides in the opposite direction. In contrast, PLTP primarily mediates the transfer of phospholipids between lipoproteins [14,15]. These processes contribute to HDL size heterogeneity and are key mechanisms in the redistribution of cholesterol among lipoproteins, also facilitating hepatic cholesterol uptake via the LDL receptor. Additionally, cholesterol from HDL is returned to the liver through other pathways, such as via the hepatic scavenger receptor class B type 1 (SR-B1), enabling the liver to recycle or excrete cholesterol [16].

Variants in genes associated with reverse cholesterol transport play a significant role in determining HDL-C levels. These genes influence various stages of HDL metabolism, and extreme HDL-C levels have been linked to notable cardiovascular health implications in affected individuals [9,17,18]. In Norway, patients with extreme HDL-C levels (≤ 0.5 mmol/L or ≥ 3 mmol/L) have been offered genetic testing of the

five key genes *ABCA1, APOA1, CETP, LCAT* and *SCARB1* (encoding SR-B1). At the Unit for Cardiac and Cardiovascular Genetics, Oslo University Hospital, approximately 270 individuals have been assessed over the past 25 years, primarily referred to us due to personal or family history of heart disease or dyslipidemia. In comparison, more than 70 000 patients with abnormal LDL cholesterol (LDL-C) levels have been analyzed in the same period. Patient numbers were based on data from our diagnostic clinic that have not been previously published. While familial hypercholesterolemia is estimated to affect approximately 1 in 300 individuals in the general population, genetic testing for HDL-related disorders remains rare, resulting in an unclear understanding of the prevalence of inherited HDL deficiencies [19]. Further, despite extensive population studies on HDL-related genes, functional analyses of identified variants are limited, leaving a gap in our understanding of the genotype-phenotype relationship [20].

Due to infrequent clinical referrals of HDL-related dyslipidemia and potentially under-examined phenotype caused by the rarity of individuals with extreme HDL-C levels, we have in this study utilized one of the large Norwegian health surveys to evaluate the health care offered to these patients in Norway. In collaboration with the Tromsø Study, we identified individuals with HDL-C ≤ 0.5 mmol/L or ≥ 3 mmol/L, and analyzed *ABCA1, APOA1, CETP, LCAT, PLTP* and *SCARB1* for disease causing variants in these individuals. While *PLTP* is not part of the current diagnostic gene panel at our unit, it was included because the same experimental system used for functional characterization of *CETP* variants was applicable to PLTP, allowing parallel assessment. Given the limited functional data available for PLTP and its established role in HDL metabolism, we considered this an opportunity to explore the contribution of this gene to extreme HDL-C phenotypes. Further, we assessed all variants found according to the guidelines from the American College of Medical Genetics and Genomics and The Association for Molecular Pathology (ACMG/AMP; hereafter referred to as ACMG) [21,22], applying functional protein-specific characterization and clinical data from the UK Biobank. While the findings of this study reinforce the rarity of monogenic causes of extreme HDL-C levels, they have also led us to modify the criteria for genetic testing of HDL-related dyslipidemia to alleviate sex discrimination in our clinic.

## Materials and methods

### Study population and design

The Tromsø Study, established in 1974, is an ongoing population-based multipurpose health study of adults in the municipality of Tromsø in Northern Norway [23]. Seven data collections have been implemented between 1974 and 2016, and the eighth wave of the Tromsø Study is ongoing. Due to limited biological material available from the first three surveys, we used data from the fourth (Tromsø-4, 1994–1995, n = 27 158), fifth (Tromsø-5, 2001, n = 8 130), sixth (Tromsø-6, 2007–2008, n = 12 984) and seventh survey (Tromsø-7, 2015–2016, n = 21 083) for this project. The participant inclusion criteria were HDL-C levels ≤ 0.5 mmol/L or ≥ 3.0 mmol/L in at least one participation. These stringent thresholds were historically selected in clinical settings to enrich for individuals with a high likelihood of monogenic causes of extreme HDL-C levels, at a time when sequencing capacity and costs limited broader genetic screening. Importantly, individuals with known monogenic disorders affecting HDL metabolism have been shown to present with HDL-C levels within these ranges, supporting the biological relevance of the applied cutoffs [24–26]. Cholesterol levels from all participations were obtained to monitor the normal fluctuation in individual lipid profiles. The biological material obtained was DNA, and the extension of the sequencing analyses performed was dependent on the availability of the biological material.

### Authorizations

This research project was approved by the Tromsø study Data and Publication Committee (Project 8030.00457) and by the Regional Committee for Medical Research Ethics – South East (Application 142265). The study was conducted according to the Declaration of Helsinki. Written informed consent was collected by the Tromsø Study from all participants. The samples and data were accessed for this research project on 03.05.2022.

## Numbering of nucleotides and codons

The following transcripts were used: *ABCA1*, NM_005502.4; *APOA1*, NM_000039.3; *CETP*, NM_000078.3; *LCAT*, NM_000229.2; *PLTP*, NM_006227.4; *SCARB1*, NM_005505.5. Codon number 1 was defined as the ATG start codon, with the A of the start codon set as nucleotide number 1. The genetic variants at protein level are noted with prefix p. throughout the manuscript.

## UK Biobank study and whole exome sequencing data

The UK Biobank study, described in detail previously, is a population-based prospective cohort conducted in the United Kingdom in which > 500 000 individuals aged 40–69 years were included from 2006 to 2010 [27]. The study has been approved by the North West Multi-Centre Research Ethics Committee for the United Kingdom, from the National Information Governance Board for Health and Social Care for England and Wales, and by the Community Health Index Advisory Group for Scotland. The present research has been conducted using the UKBB resource under the application number 49823. The records of individuals who have withdrawn from UKBB were removed from the analyses. Genetic variants in genes of interest were screened using whole-exome sequencing (WES) data from the UK Biobank. WES sequencing technical and analytic details have been reported previously [28]. In short, WES was performed using IDT xGen Exome Research Panel v1.0, targeting 38 997 831 bases in 19 396 genes. Exomes were captured including 100 bp flanking regions. Among participants we focused on 469 835 individuals for whom WES data was available. Variant filtering (GRCh38) was performed using vcf files (format VCFv4.2) with bcftools (v1.14) and jvarkit [29,30]. Genetic and phenotypic data were combined and processed using RStudio (v.2022.02.1).

## DNA sequencing and variant classification

DNA was analyzed by standardized accredited Sanger sequencing. Oligonucleotide sequences used for PCR amplification are available upon request. Genetic variants were assessed according to the ACMG guidelines, categorizing variants based on their pathogenic potential as benign (class 1), likely benign (class 2), uncertain significance (class 3), likely pathogenic (class 4) or pathogenic (class 5) [21,31]. The filtering allele frequency cutoffs for the benign criteria BA1 and BS1 were set at ≥ 0.005 and ≥ 0.002, respectively, while for the pathogenic PM2 criterion, the maximum allele frequency cutoff was ≤ 0.0002 [32]. For *CETP* variants with the highest allele frequency in the East Asian population, the second highest allele frequency among the populations in the Genome Aggregation Database (gnomAD v.4.1.0) was evaluated for the BA1/BS1 criteria, due to the frequency of increased HDL-C in the Japanese population [33]. The pathogenic phenotype criterion PP4 was applied as strong evidence when variants were identified in individuals with HDL-C ≤ 0.1 mmol/L or with Tangier disease, LCAT deficiency or Fish-eye disease, or for loss-of-function characterization in patient material from the literature.

## Functional assays and distribution of HDL-C levels

Methods and results from HDL-C distribution and functional assays characterizing variants in *ABCA1*, *CETP*, *LCAT* and *SCARB1* can be found in S1 File.

## Statistical analyses

Data are presented as mean (± standard deviations (SD)) unless otherwise stated. GraphPad Prism 10.1.2 (GraphPad Software LLC, Boston, MA) was used to calculate *p* values by an ordinary one-way ANOVA followed by Dunnett's multiple comparisons test. Effect sizes were calculated using Hedges' *g* in Microsoft Excel. A positive value of *g* indicates higher activity in the variant compared to the wild-type, while a negative value indicates reduced activity. A significance level of $p < 0.05$ was used.

## Results

### Participants in the Tromsø Study with HDL-C levels ≤ 0.5 mmol/L or ≥ 3 mmol/L

In line with current thresholds for HDL-related genetic testing in Norway, we screened the Tromsø Study for participants with HDL-C levels at the extreme ends of the distribution. Across the Tromsø-4 to Tromsø-7 surveys, conducted between 1994 and 2016, a total of 36 626 unique individuals participated. Of these, 210 participants (0.57%) had at least one recorded HDL-C measurement of ≤ 0.5 mmol/L or ≥ 3 mmol/L (Fig 1). Specifically, 198 individuals (0.54%) had HDL-C levels exceeding 3 mmol/L (high HDL-C group), while 12 individuals (0.033%) had HDL-C levels of 0.5 mmol/L or lower (low HDL-C group). Several participants were involved in multiple surveys, and 165 of the 210 individuals had more than one HDL-C measurement available. On group level, utilizing all available measurements for each individual, the average HDL-C levels for the low and high HDL-C groups were 0.57 mmol/L and 2.87 mmol/L, respectively.

While individual lipid profiles typically fluctuate, dyslipidemia caused by monogenic factors generally results in a consistent abnormal lipid profile across multiple measurements over time [34]. The average HDL-C levels for the included participants varied significantly, particularly for the high HDL-C group (S1 Table). Furthermore, the proportion of individuals with high HDL-C levels (≥ 3 mmol/L) varied across the four surveys: 0.13% in Tromsø-4 (35/27 158), 0.17% in Tromsø-5 (14/8 130), 0.32% in Tromsø-6 (42/12 984), and 0.65% in Tromsø-7 (137/21 083). This represents a fivefold increase in the proportion of individuals with extremely high HDL-C levels in Tromsø-7 compared to Tromsø-4, and roughly double the proportion seen in Tromsø-6. Whether this variation is attributable to demographic changes such as increasing participant age in later surveys, or due to systematic differences in HDL-C measurement methods over time, remains uncertain.

While genetic dyslipidemia is primarily autosomally inherited, a sex segregation between the two groups was suspected based on sex-adjusted HDL-C reference limits. Accordingly, all but one of the individuals with low HDL-C levels were men, while 83% (164 of 198) in the high HDL-C group were women. This sex discrimination is in accordance with the findings in a similar study by Sadananda et al. of 80% and 26% male in low and high HDL-C group, respectively [35].

Sex-specific distributions of HDL-C in a large clinical dataset further demonstrated marked differences between men and women (S1 Fig). Using current extreme HDL-C thresholds (≤ 0.5 mmol/L and ≥ 3.0 mmol/L), only a very small fraction of the population was captured, with clear sex imbalance. Among men, 0.03% had HDL-C ≤ 0.5 mmol/L and 0.03% had HDL-C ≥ 3.0 mmol/L, whereas among women, 0.01% had HDL-C ≤ 0.5 mmol/L and 0.15% had HDL-C ≥ 3.0 mmol/L.

### Variants of interest identified in subjects at the extreme ends of the HDL-C distribution

DNA from all 210 participants in both the high and low HDL-C groups were sequenced for variants in six major HDL-related genes: *ABCA1*, *APOA1, CETP*, *LCAT*, *PLTP* and *SCARB1*. In total, 38 heterozygous distinct variants of interest including all missense and promoter variants, as well as synonymous and intronic variants predicted to affect mRNA splicing, were found across the six genes: *ABCA1* (n = 12), *APOA1* (n = 4), *CETP* (n = 11), *LCAT* (n = 3), *PLTP* (n = 3) and *SCARB1* (n = 5) (Table 1). A complete list of additional variants detected in the cohort is available in S2 and S3 Tables.

For missense and splice variants of interest, functional characterization was performed utilizing protein specific activity assays (ABCA1, CETP, LCAT and SCARB1) or minigene assay, respectively (S2–S5 Figs, Table 1). Additionally, qPCR analyses showed no differences in transfection efficiency between constructs (S6 Fig). Most tested variants demonstrated functional activity comparable to wild-type, consistent with benign or likely benign classification. In contrast, marked loss of function was observed for a limited number of variants. These included the novel ABCA1 p.G1818E variant, the CETP variants p.Q337X, p.L290P, p.D459G and the splice-site variant c.1321 + 1G > A, the LCAT p.M276K variant, and the synonymous SCARB1 variant c.591C > T (p.G197G). In addition, population-level data from the UK Biobank were used to assess average HDL-C levels in variant carriers compared with non-carriers, supporting the functional interpretation and classification of variants (Table 1).

| Tromsø Study Stats | | | | |
|---|---|---|---|---|
| | Tromsø 4 (1994-1995) | Tromsø 5 (2001) | Tromsø 6 (2007-2008) | Tromsø 7 (2015-2016) |
| **Participants** | 27 158 | 8 130 | 12 984 | 21 083 |
| **Male** | 12 856 | 3 511 | 6 054 | 10 009 |
| **Female** | 14 293 | 4 619 | 6 930 | 11 074 |
| **Age range** | 25 - 97 | 30 - 89 | 30 - 89 | 40 - 99 |

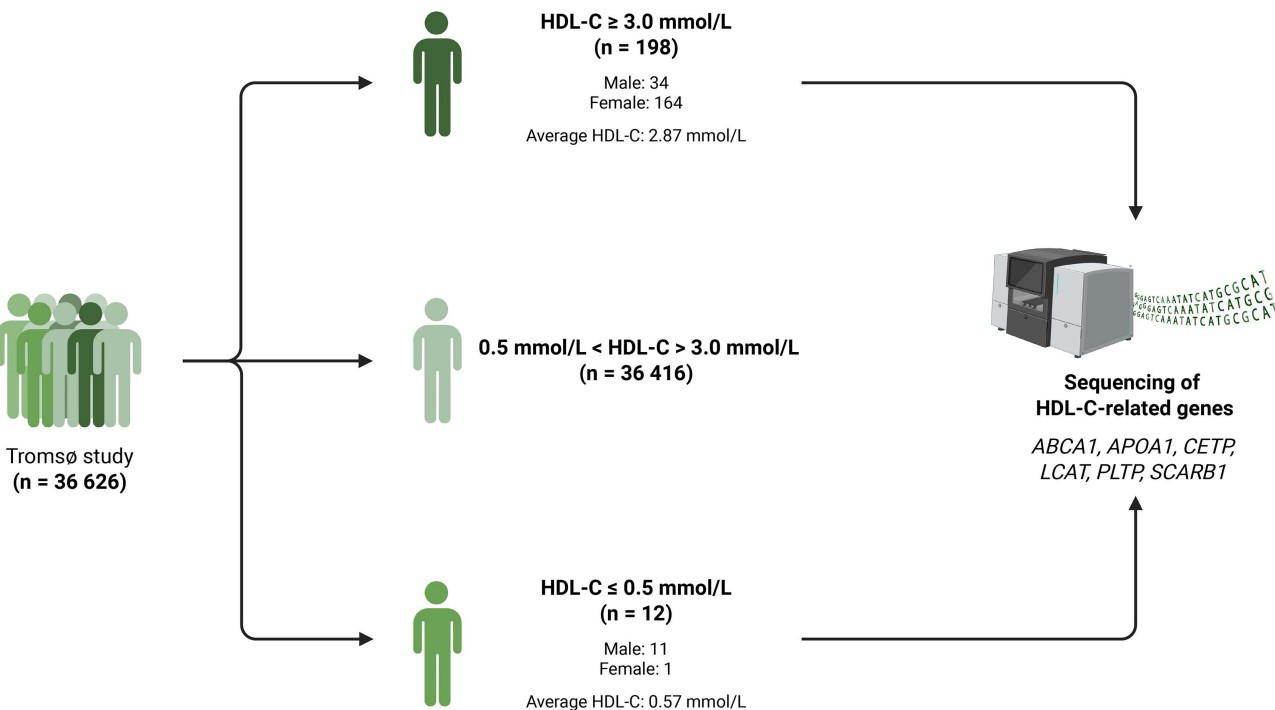

**Fig 1. Overview of the four Tromsø Studies included in this article.** The figure depicts the number of participants, sex distribution, and age range for each survey. Across all four studies, a total of 36 626 unique individuals were included. Among these, only 210 participants had at least one extreme HDL-C measurement – defined as ≤ 0.5 mmol/L (n = 12) or ≥ 3.0 mmol/L (n = 198). Participants were sequenced in six HDL-related genes (*ABCA1, APOA1, CETP, LCAT, PLTP, SCARB1*). Created using BioRender.

Variants found in both extreme ends of the lipid profile spectrum are less likely to be the primary genetic cause of the phenotype, thus eleven variants were classified as benign (class 1). Furthermore, seven variants were classified as likely benign (class 2), resulting in a total of 18 variants (47%) classified as not causative for the extreme HDL-C phenotype. This classification is in addition based on the high allele frequencies of these variants in the general population, in many cases their low evolutionary conservation across species, and evidence of normal protein activity in functional assays (S2–S6 Figs). Across all six genes, a total of 14 variants (37%) were assessed as variants of uncertain significance (class

**Table 1. Variants of interest identified in extreme HDL-C phenotype.**

| Protein | Nucleotide | Phenotype Tromsø [HDL-C] n (%) | | Functional assay | | ACMG Class | Relative HDL-C UK biobank % (n) |
|---|---|---|---|---|---|---|---|
| | | High | Low | % of WT (SD) | References | | |
| **ABCA1** | | | | | | | |
| p.R219K | c.656G>A | 86 (43.4%) | 6 (50%) | 90% (± 7) | [36] | 1 | −0.21% (195 930) |
| p.S296T | c.886T>A | 1 (0.5%) | – | 98% (± 11) | S2 Fig | 2 | −4.63% (18) |
| p.V771M | c.2311G>A | 11 (11.1%) | 2 (16.7%) | 85% (± 14) | [36] | 1 | 2.39% (24 913) |
| p.V825I | c.2473G>A | 25 (12.6%) | 2 (16.7%) | 97% (± 7) | S2 Fig | 1 | 1.62% (50 059) |
| p.I883M | c.2649A>G | 37 (18.7%) | 2 (16.7%) | 84% (± 8) | [37] | 1 | 1.18% (104 371) |
| p.C887F | c.2660G>T | 2 (1%) | – | 87% (± 9) | [37] | 1 | −4.12% (66) |
| p.E1172D | c.3516G>C | 12 (6.1%) | 1 (8.3%) | 74% (± 9) | [37] | 1 | 1.37% (26 132) |
| p.S1181F | c.3542C>T | 1 (0.5%) | – | 69% (± 19) | [37] | 3 | −4.97% (1429) |
| p.K1587R | c.4760A>G | 132 (66.7%) | 4 (33.3%) | 106% (± 5) | [36] | 1 | 1.66% (380 842) |
| p.V1674I | c.5020G>A | 2 (1%) | – | 77% (± 10) | S2 Fig | 1 | −2.20% (111) |
| p.G1818E | c.5453G>A | – | 2 (16.7%) | 31% (± 10) | S2 Fig | 4 | −32.75% (4) |
| p.R1925Q | c.5774G>A | 4 (2%) | – | 100% (± 12) | [37] | 2 | 4.05% (547) |
| **CETP** | | | | | | | |
| p.A15G | c.44C>G | 3 (1.5%) | – | 121% (± 23) | [38] | 1 | 8.94% (1481) |
| p.D131N | c.391G>A | 1 (0.5%) | – | 64% (± 23) | [38] | 3 | 11.43% (61) |
| p.L290P | c.869T>C | 1 (0.5%) | – | 3% (± 4) | S3 Fig | 3 | 20.72% (182) |
| p.Q337X | c.1009C>T | 4 (2%) | – | 2% (± 3) | S3 Fig | 5 | −16.01% (1) |
| p.V385M | c.1153G>A | 1 (0.5%) | – | 81% (± 12) | S3 Fig | 2 | 0.05% (1296) |
| p.A390P | c.1168G>C | 9 (4.5%) | 1 (8.3%) | 228% (± 50) | [38] | 1 | −6.84% (42 886) |
| p.V422I | c.1264G>A | 171 (86.4%) | 8 (66.7%) | 110% (± 25) | [38] | 1 | – |
| p.E443K | c.1327G>A | 1 (0.5%) | – | 18% (± 6) | [38] | 3 | 25.27% (18) |
| p.D459G | c.1376A>G | 1 (0.5%) | – | 19% (± 11) | [38] | 5 | 7.75% (158) |
| p.R468Q | c.1403G>A | 3 (1.5%) | 1 (8.3%) | 73% (± 9) | [38] | 1 | −7.22% (30 414) |
| Intron | c.1321+1G>A | 1 (0.5%) | – | r.1249_1321del, fsTer4[a] | S3 Fig | 5 | 25.52% (29) |
| **LCAT** | | | | | | | |
| p.S232T | c.694T>A | 7 (3.5%) | 3 (25%) | 84% (± 9) | S4 Fig | 1 | −1.78% (26 000) |
| p.M276K | c.827T>A | – | 2 (16.7%) | 2% (± 3) | S4 Fig | 5 | – |
| p.E378K | c.1132G>A | 1 (0.5%) | – | 57% (± 6) | S4 Fig | 3 | −4.55% (164) |
| **SCARB1** | | | | | | | |
| p.G2S | c.4G>A | 28 (14.1%) | 1 (8.3%) | 103% (± 7) | S5 Fig | 1 | −0.23% (76 978) |
| p.G12R | c.34G>A | 1 (0.5%) | – | 100% (± 7) | S5 Fig | 3 | −0.89% (1) |
| p.V135I | c.403G>A | 1 (0.5%) | – | 61% (± 7) | S5 Fig | 1 | −1.24% (9667) |
| p.I231V | c.691A>G | 1 (0.5%) | – | 69% (± 8) | S5 Fig | 3 | – |
| Silent | c.591C>T | 1 (0.5%) | – | r.590_630del, fsTer34[a] | S5 Fig | 3 | – |
| **APOA1** | | | | | | | |
| Promoter | c.-12C>T | – | 1 (8.3%) | – | – | 3 | – |
| p.E100Q | c.298G>C | 1 (0.5%) | – | – | – | 3 | 4.09% (2) |
| p.R184L | c.551G>T | – | 1 (8.3%) | – | – | 4 | – |
| p.A188S | c.562G>T | 2 (1%) | – | – | – | 3 | −0.72% (849) |

*(Continued)*

**Table 1.** (Continued)

| Protein | Nucleotide | Phenotype Tromsø [HDL-C] n (%) | | Functional assay | | ACMG Class | Relative HDL-C UK biobank % (n) |
|---|---|---|---|---|---|---|---|
| | | High | Low | % of WT (SD) | References | | |
| **PLTP** | | | | | | | |
| p.E211Q | c.631G>C | 3 (1.5%) | – | – | – | 3 | 9.66% (69) |
| p.R380W | c.1138C>T | 2 (1%) | – | – | – | 3 | 0.69% (867) |
| p.V422M | c.1264G>A | 2 (1%) | – | – | – | 3 | – |

Missense and promoter variants, as well as synonym and intronic variants predicted by SpliceAI to affect mRNA splicing, in the genes *ABCA1*, *APOA1*, *CETP*, *LCAT*, *PLTP* and *SCARB1* from 210 participants with extreme HDL-cholesterol (HDL-C) levels in the Tromsø Study. All variants found are listed in S2 and S3 Tables. For each variant, the table provides the amino acid change (missense variants), nucleotide change, the number and percentage of carriers in the high and low HDL-C groups from the Tromsø Study (Phenotype Tromsø [HDL-C]), results from functional assays compared to wild-type, and references to where the assays were performed (S2–S6 Figs). Variants are classified according to the guidelines from The American College of Medical Genetics and Genomics and The Association for Molecular Pathology (ACMG) as class 1 (benign), class 2 (likely benign), class 3 (variant of uncertain significance), class 4 (likely pathogenic), or class 5 (pathogenic), with classification criteria outlined in S2 and S3 Tables. The final column includes UK Biobank data, showing the number of individuals carrying each variant and the corresponding percentile change in HDL-C levels relative to the population average.

SD: standard deviation. [a]Functional activity given as effect of variant on RNA level and position of frameshift-induced stop codon.

3), lacking sufficient data to establish pathogenic potential. Noteworthy, although the number of variants found in *ABCA1* and *CETP* exceeds the total found in the other four genes, the low frequency of class 3 variants in *ABCA1* (1 of 12) and *CETP* (3 of 11) is a result of three recent studies on functional variant assessment [36–38].

Only six variants were classified as likely pathogenic (class 4, n = 2) or pathogenic (class 5, n = 4). However, four additional variants of uncertain significance (class 3) showed evidence from functional assays and data from the UK Biobank suggesting that they may contribute to the HDL-C phenotype.

## Variants contributing to abnormal HDL-C levels

Among individuals with HDL-C ≤ 0.5 mmol/L, we identified three pathogenic missense variants in five occurrences across four individuals: *ABCA1* p.G1818E (n = 2), *LCAT* p.M276K (n = 2) and *APOA1* p.R184L (n = 1), including one individual harboring both *ABCA1* p.G1818E and *LCAT* p.M276K. While the *APOA1* p.R184L and *LCAT* p.M276K have previously been associated with HDL deficiency [39,40], the *ABCA1* p.G1818E variant represents a novel finding. Carriers of *ABCA1* p.G1818E in the UK Biobank (n = 4) had in average a 32.75% lower HDL-C level compared to non-carriers. This variant also demonstrated only 31% of wild-type cholesterol efflux activity in functional assays (S2 Fig). Together, these findings provide sufficient evidence to classify the variant as likely pathogenic (class 4).

In the high HDL-C group, we identified three *CETP* variants with strong evidence for pathogenicity in six individuals: p.Q337X (n = 4), p.D459G (n = 1) and c.1321 + 1G > A (n = 1) (S3 Fig). Although the p.Q337X variant was reported in only one individual with reduced HDL-C compared to non-carriers in the UK Biobank data, all three variants were classified as class 5 (pathogenic) according to the ACMG guidelines.

Additionally, we identified four variants of uncertain significance that may contribute to elevated HDL-C levels. These include three *CETP* variants (p.D131N, p.L290P and p.E443K) and one synonymous variant in *SCARB1* (c.591C > T; p.G197G) predicted to disrupt splicing. The *CETP* variants p.D131N, p.L290P, and p.E443K had multiple carriers in the UK Biobank and were associated with notable increases in HDL-C compared to non-carriers – 11.43%, 20.72%, and 25.27%, respectively, in addition to reduced protein functionality (S3 Fig). The *SCARB1* p.G197G was not present in the UK Biobank data but induced an out-of-frame deletion of 41 nucleotides in exon 4 of *SCARB1* (r.590_630del) when analyzed by minigene assay (S5 Fig). In coherence with the SpliceAI-predicted donor gain, this results in a premature stop

codon early in *SCARB1* exon 6, indicative of nonsense mediated decay degradation of the affected allele. The functional analyses of all four variants combined with elevated average HDL-C levels in carriers from the UK Biobank for the *CETP* variants support their potential pathogenicity.

Likely genetic causes were identified in 4 of 12 individuals (33.3%) in the low HDL-C group and in 10 of 198 individuals (5.05%) in the high HDL-C group (Table 2), in accordance with Sadananda et al. (35.9% in low HDL-C, 5.2% in high HDL-C) [35]. The average HDL-C level among individuals with likely pathogenic or pathogenic variants in the low HDL-C group was 0.53 mmol/L, slightly lower than the overall group average of 0.57 mmol/L. In the high HDL-C group, the corresponding average was 2.96 mmol/L, slightly higher than the group average of 2.87 mmol/L. Notably, when considering only the six individuals with variants classified as pathogenic (class 5), the average HDL-C level exceeded the study inclusion criteria, at 3.08 mmol/L. For clinical context, S4 Table summarizes variants previously identified at our diagnostic unit that are considered likely causes of abnormal HDL-C levels.

## Discussion

### Historical perspective and rationale for studying extreme HDL-C phenotypes

Individuals with extremely abnormal lipid levels are significantly more likely to carry rare, monogenic variants than those with values closer to the population mean [41,42]. Genetic testing of HDL-related dyslipidemia has thus primarily been reserved for individuals with HDL-C values at the extreme ends of the distribution. In our laboratory, these thresholds were defined as ≤ 0.5 mmol/L or ≥ 3.0 mmol/L – values that are exceptionally rare in the general population [39]. Since

**Table 2. Individuals with potentially causative variants of extreme HDL-C levels.**

| Individual | Variant (Gene) | Sex | Age group | HDL-C | | | |
|---|---|---|---|---|---|---|---|
| | | | | Tromsø 4 (1994-1995) | Tromsø 5 (2001) | Tromsø 6 (2007-2008) | Tromsø 7 (2015-2016) |
| **Low HDL-C** | | | | | | | |
| 1 | R184L (*APOA1*) | Male | 50–59 | 0.36 | | | |
| 4 | G1818E (*ABCA1*) | Male | 30–39 | 0.54 | | | 0.40 |
| 6 | G1818E (*ABCA1*)/M276K (*LCAT*) | Male | 50–59 | 0.63 | 0.47 | | |
| 10 | M276K (*LCAT*) | Male | 20–29 | 0.48 | | 0.70 | 0.68 |
| **High HDL-C** | | | | | | | |
| 28 | D131N (*CETP*) | Male | 50–59 | 1.73 | 1.97 | 3.30 | 2.80 |
| 100 | E443K (*CETP*) | Female | 30–39 | 2.66 | | | 3.10 |
| 102 | Q337X (*CETP*) | Female | 30–39 | 2.04 | | 3.20 | 3.40 |
| 110 | Q337X (*CETP*) | Male | 60–69 | 2.77 | 2.66 | 3.30 | |
| 141 | L290P (*CETP*) | Female | 50–59 | 3.16 | | 3.00 | |
| 142 | Q337X (*CETP*) | Female | 40–49 | 2.35 | | 3.10 | 3.80 |
| 166 | c.1321+1G>A (*CETP*) | Female | 60–69 | | | 2.80 | 3.60 |
| 192 | c.591C>T (*SCARB1*) | Female | 40–49 | | | | 3.40 |
| 194 | Q337X (*CETP*) | Female | 40–49 | | | | 3.40 |
| 200 | D459G (*CETP*) | Female | 50 - 59 | | | | 3.60 |

Four out of 12 individuals (33.33%) in the low HDL-C group (HDL-C ≤ 0.5 mmol/L) and 10 out of 198 individuals (5.05%) in the high HDL-C group (HDL-C ≥ 3.0 mmol/L) had genetic variants which could potentially explain their extreme HDL-C levels. The table displays an anonymous individual code, the identified variant along with the corresponding gene, sex and age group of the participants at first entry, and HDL-cholesterol (HDL-C) levels across the Tromsø Studies.

2007, approximately 270 individuals have been tested – most referred due to a personal or family history of cardiovascular disease or dyslipidemia in combination with markedly abnormal HDL-C levels. To date, the unit has identified 41 likely genetic causes (25 variants) of abnormal HDL-C levels (S4 Table), corresponding to 15.19% of all referred individuals.

Given the growing body of evidence that both very low and very high HDL-C levels may be associated with increased overall mortality, the low frequency of HDL-related referrals to our clinic and the relatively high hit rate of pathogenic variants identified, we aimed to evaluate the prevalence of pathogenic variants in individuals with extreme HDL-C levels in a population-based setting. Specifically, we assessed some of the most prominent HDL-associated genes – *ABCA1*, *APOA1*, *CETP*, *LCAT*, *PLTP* and *SCARB1*. Through collaboration with the Tromsø Study, we identified 210 individuals eligible for genetic testing by the current HDL-C criteria from a cohort of 36 626 participants.

## Genetic variants associated with abnormal HDL-C levels in study participants

Among the 210 individuals with abnormal HDL-C levels, 14 carried variants likely to explain their HDL-C phenotypes. Six of these variants were classified as pathogenic or likely pathogenic according to ACMG criteria, including *LCAT* p.M276K, *APOA1* p.R184L and *CETP* p.Q337X, p.D459G and c.1321+1G>A. Additionally, the *ABCA1* p.G1818E variant was classified as likely pathogenic when incorporating functional characterization from this study.

Several variants of uncertain significance appear to be strong candidates for pathogenicity based on *in vitro* functional data and population-level associations. The three *CETP* variants p.D131N, p.L290P and p.E443K all demonstrate reduced CETP function in vitro. Of these, p.L290P and p.E443K retained only 3% and 18% of wild-type CETP activity, respectively. Both variants have been identified in patients at our clinic with relatively high HDL-C: p.E443K in a male with HDL-C>3 mmol/L and p.L290P in a female with HDL-C>2.5 mmol/L. Support for their pathogenicity is strengthened by UK Biobank data, where carriers of p.E443K (n=18) show an average HDL-C increase of 25.27% compared to non-carriers, while p.L290P showed even greater functional impairment which is supported by data from 182 carriers with an average HDL-C increase of 20.72%. Notably, p.E443K is one moderate ACMG criterion short of being reclassified as likely pathogenic (class 4). Although p.L290P currently fulfills fewer ACMG criteria than p.E443K, its combined functional and population-level data strongly support a causal role in elevated HDL-C levels.

The variant for which we have the least confidence, yet still consider potentially contributory, is *CETP* p.D131N. Functional assays indicated that this variant retains 64% of wild-type activity, suggesting a moderate reduction in function. However, UK Biobank data show that 61 carriers of p.D131N have an average HDL-C increase of 11.43%, exceeding the 7.75% increase observed in carriers of the pathogenic variant p.D459G (n=158). Taken together, these findings support p.E443K and p.L290P as likely contributors to elevated HDL-C, while suggesting that p.D131N may exert a more modest but clinically relevant effect.

Koenig et al. recently identified the first reported *SCARB1* null variant c.754_755delinsC together with p.G319V in compound heterozygote siblings with early-onset severe coronary artery disease [43]. The c.754_755delinsC variant introduces a frameshift-induced premature stop codon at p.253, resulting in minimal RNA and no detectable protein expression. The *SCARB1* p.G197G variant identified in one Tromsø Study participant activates an alternative splice donor in exon 4, leading to a frameshift deletion and a premature stop codon at p.243. Although p.G197G does not meet ACMG criteria for likely pathogenicity, functional data reported by Koenig et al. indicate that a truncation at p.243 is also consistent with a *SCARB1* null variant. Notably, both heterozygote and compound heterozygote cases in Koenig et al. had HDL-C levels within the normal range, underscoring a role for SR-B1 in severe atherosclerotic disease beyond extreme HDL-C levels.

## Establishing new HDL-C thresholds for genetic testing: balancing prevalence, risk and yield

Despite extensive experience with genetic testing for elevated LDL-C at our diagnostic unit, amounting to nearly 70 000 individuals, testing for HDL-related disorders has historically remained limited. Several factors may contribute to this: the

small number of known pathogenic variants, the perception that monogenic HDL disorders are exceedingly rare, limited clinical awareness, and variability in guideline recommendations. In addition, HDL-C has traditionally been regarded as the "good cholesterol", primarily due to its inverse epidemiological association with CVD risk.

In recent years, this interpretation has been challenged. Observational studies have repeatedly demonstrated a U-shaped association between HDL-C and all-cause mortality, indicating that both very low and very high HDL-C levels are associated with increased risk. Madsen et al. showed that men with HDL-C ≥ 3.0 mmol/L had a twofold higher risk of all-cause mortality, while women with HDL-C ≥ 3.5 mmol/L had a 68% increased risk. Likewise, HDL-C < 1.0 mmol/L was associated with substantially higher mortality in both sexes. Moreover, Mendelian randomization studies and randomized trials have shown that altering HDL-C concentration itself does not reduce cardiovascular events, suggesting that HDL-C is more a biomarker of metabolic status than a direct causal factor for CVD [44]. Therapeutic strategies targeting CETP, ApoA1, ABCA1 and LCAT have increased HDL-C levels but have not led to improved cardiovascular outcomes [45–47].

HDL-related genetic disorders also have implications extending beyond lipid metabolism and CVD. Variants in *ABCA1* and *CETP* have been linked to Alzheimer's disease, and *APOA1* variants to amyloidosis [48–50]. While monogenic causes of extreme HDL-C levels are rare, the prevalence and clinical impact of more common polygenic variation remain poorly understood. However, recent work using polygenic trait scores has shown that an excess burden of common HDL-associated variants can account for a substantial proportion of individuals with extreme HDL-C levels in referral cohorts [51]. As interest in HDL biology has renewed, accurate and equitable criteria for identifying individuals who may benefit from genetic testing have become increasingly important.

Clinical eligibility for HDL-related genetic testing has been based on strict, sex-neutral HDL-C thresholds (≤ 0.5 mmol/L or ≥ 3.0 mmol/L) at our diagnostic unit. In this study, such uniform cutoffs resulted in pronounced sex imbalance, with very low HDL-C occurring almost exclusively in men and very high HDL-C predominantly in women. This reflects well-established sex differences in HDL-C distribution and indicates that the previous criteria systematically biased access to genetic testing.

Population distribution data further showed that these thresholds were overly restrictive, identifying only a small fraction of individuals with extreme HDL-C (S1 Fig). Among women, approximately 0.15% had HDL-C ≥ 3.0 mmol/L, with a diagnostic yield of likely pathogenic variants comparable to that observed in other genetic testing contexts (4.88%) [52–54], supporting retention of this upper threshold. Applying a similar tail-based approach across sexes yielded revised thresholds of ≤ 0.6 mmol/L or ≥ 2.7 mmol/L for men and ≤ 0.7 mmol/L or ≥ 3.0 mmol/L for women. These sex-specific limits better reflect population distributions, reduce bias in testing eligibility, and provide a pragmatic balance between diagnostic yield and resource use, although further validation in independent cohorts is warranted.

## Variability of HDL-C levels and the importance of longitudinal data in genetic testing

Among the 210 individuals who met our inclusion criteria, most had data from multiple time points: 71 participated in two Tromsø surveys, 57 in three, and 37 in all four. When we examined the longitudinal data, we found that consistent extreme HDL-C levels were rare. In the low HDL-C group, only one individual maintained an average HDL-C level of ≤ 0.5 mmol/L across two surveys. Similarly, in the high HDL-C group, only 49 participants had a mean level of HDL-C ≥ 3.0 mmol/L or above (S1 Table).

HDL-C levels also varied over time in the participants with confirmed pathogenic variants. Each of the four individuals with low HDL-C had only one measurement ≤ 0.5 mmol/L. Even with the revised thresholds (HDL-C ≤ 0.6 mmol/L for men, HDL-C ≤ 0.7 mmol/L for women), two of the eight measurements of the four individuals would still exclude them for genetic testing. A similar pattern was observed in the high HDL-C group.

These findings highlight the substantial range of HDL-C levels in individuals with confirmed pathogenic variants, as well as the considerable intra-individual variability in HDL-C over time. Although our study lacks evidence

of increased diagnostic yield, the new revised thresholds captured more individuals with pathogenic variants in both groups, when assessing the longitudinal average HDL-C levels. This underscores the importance of repeated measurements when evaluating extreme lipid phenotypes. Individuals with consistently low or high HDL-C levels – regardless of whether they fall within the established thresholds – should thus still be considered for genetic testing of HDL-related genes.

## Limitations of study

The complex and dynamic nature of lipid metabolism complicates genotype–phenotype correlations in dyslipidemia. A key limitation of this study is therefore the restricted gene set analyzed, reflecting historical constraints of targeted genetic testing and limited DNA availability. Consequently, several HDL-associated genes beyond the canonical monogenic causes were not assessed.

Recent advances in clinical genetics now allow a more comprehensive approach. At our diagnostic unit, individuals with abnormal HDL-C are currently analyzed using whole genome sequencing (WGS) and a PanelApp-based gene panel with strong evidence of disease causation, including *LPL, ANGPTL3*, and *LIPA* [55]. Such an approach enables broader assessment of both monogenic and polygenic contributions to HDL metabolism and captures regulatory and non-coding variation [56–60]. Nevertheless, even consensus gene panels do not encompass all genes implicated in HDL remodeling and clearance. In particular, *LIPG* and *LIPC* are not included in the gene panel and were not included in the present study, despite evidence linking rare variants in these genes to extreme HDL-C phenotypes [61,62]. Their exclusion therefore represents an additional limitation and underscores the need for continued refinement of genetic testing strategies for HDL-related dyslipidemia.

Elevated triglycerides and low HDL-C are hallmark features of metabolic syndrome, familial combined hyperlipidemia, and insulin resistance [63–65]. These mechanisms suggest that individuals with both high triglycerides and low HDL-C are more likely to have secondary dyslipidemia rather than a primary genetic defect in HDL-related genes, emphasizing the importance of excluding secondary causes, before considering genetic testing for HDL disorders. Without the triglyceride data from the Tromsø Study, we cannot rule out metabolic syndrome as the cause of low HDL-C in the eight participants for whom no genetic cause was identified. Finally, we lacked systematic information on secondary causes of elevated HDL-C (e.g., alcohol intake, medications, hormonal therapy or liver disease), and cannot exclude that such factors contributed to the extreme HDL-C levels observed in some participants.

## Conclusions

Genetic testing for HDL-related dyslipidemia is underutilized, with many individuals not meeting the current extreme HDL-C thresholds. Whereas monogenic HDL disorders are rare, our study identified several pathogenic variants in individuals with extreme HDL-C levels, where functional characterization, in combination with variant-specific phenotype data from the UK Biobank, substantially aided the pathogenicity assessment. Based on our findings, we propose revised sex-specific thresholds for genetic testing: HDL-C ≤ 0.6 mmol/L or ≥ 2.7 mmol/L for men, and HDL-C ≤ 0.7 mmol/L or ≥ 3.0 mmol/L for women. These adjusted thresholds are designed to alleviate sex discrimination while keeping testing volumes manageable. As our understanding of HDL-related genetic disorders evolves, further refinement of these thresholds and the integration of functional assays will enhance the identification and management of dyslipidemia, with ongoing research playing a pivotal role in shaping future diagnostic practices.

## Supporting information

**S1 File. Supplementary material and methods.**
(PDF)

**S1 Table. An overview of all the individuals included in this study.** All individuals are indicated with sex, age group at first entry, lipid values (mmol/L) at the Tromsø 4 (1994–1995), 5 (2001), 6 (2007–2008) and 7 (2015–2016) studies and the mean HDL-cholesterol values.
(PDF)

**S2 Table. All missense variants in HDL-related genes in participants from the Tromsø Study.** Missense variants in *ABCA1* (NM_005502.4), *APOA1* (NM_000039.3), *CETP* (NM_000078.3), *LCAT* (NM_000229.2), *PLTP* (NM_006227.4) and *SCARB1* (NM_005505.5) are annotated with respect to their effects at protein and nucleotide level. The reported lipid associated phenotypic effects of the variants, as well as the classifications made by Human Gene Mutation Database (HGMD) for variants reported in that database are shown. Also shown is an *in silico* prediction of pathogenicity represented by a REVEL score [66]. Allele frequencies of the variants are obtained from the Genome Aggregation Database (gnomAD, v.4.1.0). Pathogenicity classes of the variants were assessed according to the guidelines from The American College of Medical Genetics and Genomics and The Association for Molecular Pathology (ACMG), and the criteria used are indicated [21]. The number and percentage of variant carriers among the Tromsø study participants with high or low HDL-C are also indicated. HDL-C: HDL-cholesterol level. [a]HGMD HDL-related phenotype associated with the individual variant. [b]HGMD class: DM: Disease-causing mutation; DM?: Disease-causing mutation?; DP: Disease-associated polymorphism; FP: *in vivo* or *in vitro* functional polymorphism; DFP: Disease-associated polymorphism with supporting functional evidence; -: Not listed in HGMD. [c]A higher REVEL score (from 0 to 1) indicates a greater likelihood of a deleterious variant. [d]The highest allele frequency among the populations: African/African American (AFR), Amish (AMI), Admixed American (AMR), Ashkenazi Jewish (ASJ), East Asian (EAS), Finnish (FIN), Middle Eastern (MID), Non-Finnish European (NFE) and South Asian (SAS) is shown. [e]Class 1: benign; Class 2: likely benign; Class 3: Unknown significance; Class 4: likely pathogenic; Class 5: pathogenic. [f]Criteria for pathogenicity weighed as strong (PS4, PM3_str, PP4_str), moderate (PS3_mod, PM2) or supporting (PS3–4_sup, PP3) and criteria for benignity weighed as stand-alone (BA1), strong (BS1–2), moderate (BS3_mod) or supporting (BP4).
(PDF)

**S3 Table. All silent or intronic variants in HDL-related genes in participants from the Tromsø Study.** Variants are annotated with respect to their effects at the protein and nucleotide level. The reported lipid associated phenotypic effects of the variants, as well as the classifications made by the Human Gene Mutation Database (HGMD) are shown. Also indicated are allele frequencies of the variants obtained from the Genome Aggregation Database (gnomAD, v4.1.0). Pathogenicity classes of the variants were assessed according to the guidelines from The American College of Medical Genetics and Genomics and The Association for Molecular Pathology (ACMG), and the criteria used are indicated [21]. The number and percentage of variant carriers among the Tromsø Study participants with high or low HDL-C are also indicated. HDL-C: HDL-cholesterol level. [a]HGMD reported phenotype associated with the individual variant. [b]HGMD class: DM: Disease-causing mutation; DM?: Disease-causing mutation?; DP: Disease-associated polymorphism; FP: *in vivo* or *in vitro* functional polymorphism; DFP: Disease-associated polymorphism with supporting functional evidence; -: Not listed in HGMD. [c]The highest allele frequency among the populations: African/African American (AFR), Amish (AMI), Admixed American (AMR), Ashkenazi Jewish (ASJ), East Asian (EAS), Finnish (FIN), Middle Eastern (MID), Non-Finnish European (NFE) and South Asian (SAS) is shown. [d]Class 1: benign; Class 2: likely benign; Class 3: Unknown significance; Class 4: likely pathogenic; Class 5: pathogenic. [e]Criteria for pathogenicity weighed as strong (PS4, PM3_str, PP4_str), moderate (PS3_mod, PM2) or supporting (PS3_sup, PP3–4) and criteria for benignity weighed as stand alone (BA1), strong (BS1–2), moderate (BS3_mod) or supporting (BP4, BP7). [f]SpliceAI reports Δ scores ranging from 0 to 1 reporting the probability of splice alteration induced by the variant [67].
(PDF)

**S4 Table. Genetic variants found at our unit which are likely causes of abnormal HDL-C levels.** At the Unit for Cardiac and Cardiovascular Genetics, Oslo University Hospital, we have identified 25 likely causative genetic variants in 41 of 270 individuals which have been genetically tested for abnormal HDL-C levels. The indicated HDL-C levels (mmol/L) of individuals harboring the variants are shown, along with variant pathogenicity classes and the criteria applied according to the ACMG guidelines [21]. HDL-C: HDL-cholesterol. [a]Class 1: benign; Class 2: likely benign; Class 3: Unknown significance; Class 4: likely pathogenic; Class 5: pathogenic. [b]Criteria for pathogenicity weighed as very strong (PVS1), strong (PS4, PM3_str, PP4_str), moderate (PS3_mod, PM2–3) or supporting (PS3_sup, PM3–5_sup, PP3).
(PDF)

**S1 Fig. The distribution of HDL-C.** Two histograms depicting the distribution of HDL-cholesterol (HDL-C) levels in men and women. The data were provided by the Lipid Clinic in Norway, which obtained the information from Fürst Medical Laboratory, Oslo. The dataset pertains to individuals aged 18–49.9 years, primarily measured in general practice, and therefore, the sample is not representative of the general population. A table is included to display the number and percentage of individuals with HDL-C levels above or below specific thresholds.
(PDF)

**S2 Fig. Phenotype and functional assay of variants in *ABCA1*.** Selected *ABCA1* missense variants found in the extreme ends of the HDL-C distribution in the Tromsø study were chosen for functional characterization. A) Participant HDL-C levels are given individually (female: light green; male: dark green) and as mean and standard deviation (SD) for each variant. B) Relative cholesterol efflux activity of *ABCA1* missense variants (light blue) normalized to wild-type (WT, dark blue) in transiently transfected HEK293 cells. Two loss-of-function controls (white) are shown [36]. *p* values and effect size (*g*) compared to WT are given. One representative western blot showing protein amounts in lysates from transiently transfected HEK293 cells is shown.
(PDF)

**S3 Fig. Phenotype and functional assay of variants in *CETP*.** Selected *CETP* variants found in the extreme ends of the HDL-C distribution in the Tromsø study were chosen for functional characterization. *APRQ: p.A390P and p.R468Q. A) Participant HDL-C levels are given individually (female: light green; male: dark green) and as mean (SD) for each variant. B) Relative lipid transfer activity of *CETP* missense variants (light blue) normalized to wild-type (WT, dark blue) in media from transiently transfected HEK293 cells. One loss-of-function control (white) is shown [38]. *p* values and effect size (*g*) compared to WT are given. One representative western blot showing protein amounts in media and lysates from transiently transfected HEK293 cells is displayed. C) A DNA fragment spanning intron 13 to intron 15 in *CETP* was cloned into the pET01 minigene to characterize the consequence of the variant *CETP* c.1321+1G>A. Gel electrophoresis of RT-PCR outcome from transiently transfected HEK293 cells, *in silico* prediction from SpliceAI and sequencing result (consequence) are shown. Canonical exons are shown as boxes (pET01: white; *CETP*: blue) separated by introns (black horizontal lines). Solid black lines depict normal splicing (WT), disease-associated splicing indicated by dotted lines. Red arrow indicates the approximate location of the variant.
(PDF)

**S4 Fig. Phenotype and functional assay of variants in *LCAT*.** Selected *LCAT* missense variants found in the extreme ends of the HDL-C distribution in the Tromsø study were chosen for functional characterization. A) Participant HDL-C levels are given individually (female: light green; male: dark green) and as mean (SD) for each variant. B) Relative acyltransferase activity of *LCAT* missense variants (light blue) normalized to wild-type (WT, dark blue) in media from transiently transfected HEK293 cells. One loss-of-function control (white) is shown [68]. *p* values and effect size (*g*) compared to WT are given. One representative western blot showing protein amounts in media and lysates from transiently transfected HEK293 cells is shown.
(PDF)

**S5 Fig. Phenotype and functional assay of variants in *SCARB1*.** Selected *SCARB1* variants found in the extreme ends of the HDL-C distribution in the Tromsø study were chosen for functional characterization. A) Participant HDL-C levels are given individually (female: light green; male: dark green) and as mean (SD) for each variant. B) Relative HDL-binding and uptake for *SCARB1* missense variants (light blue) normalized to wild-type (WT, dark blue) in transiently transfected HEK293 cells. One loss-of-function control (white) is shown [69]. *p* values and effect size (*g*) compared to WT are given. One representative western blot showing protein amounts in media and lysates from transiently transfected HEK293 cells is displayed. C) A DNA fragment spanning intron 3 to intron 4 in *SCARB1* was cloned into the pET01 mini-gene to characterize the consequence of the variant *SCARB1* c.591C > T (p.G197G). Gel electrophoresis of RT-PCR out-come from transiently transfected HEK293 cells, *in silico* prediction from SpliceAI and sequencing result (consequence) are shown. Canonical exons are shown as boxes (pET01: white; *SCARB1*: blue) separated by introns (black horizontal lines). Solid black lines depict normal splicing (WT), disease-associated splicing indicated by dotted lines. Red arrow indi-cates the approximate location of the variant.
(PDF)

**S6 Fig. mRNA expression.** RNA was isolated from HEK293 cells transiently transfected with *ABCA1*, *CETP*, *LCAT* or *SCARB1* wild-type (WT, dark blue columns), missense variants (light blue columns) and negative control variants (white columns). RNA was transcribed to cDNA, which was analyzed using PrimeTime Predesigned qPCR Assay primers. mRNA amounts were determined and normalized to the housekeeping gene *GAPDH* by the $2^{-\Delta\Delta Ct}$ method [70] and normalized to WT in three independent experiments. Error bars represent 1 SD. *APRQ: p.A390P and p.R468Q.
(PDF)

**S1 Raw Images. Raw blot and gel images.**
(PDF)

**S1 Raw Data. Raw data.**
(PDF)

## Acknowledgments

We thank the Norwegian Health Association for their interest and engagement in this research. We would also like to thank the Tromsø Study for granting access to data and DNA samples from the 210 individuals with extreme HDL-C lev-els. Our appreciation extends to Kjetil Retterstøl and the Lipid Clinic in Norway, as well as to Fürst Medical Laboratory and its director, Håvard Selby Ebbestad, for providing the HDL-C distribution data. The present research has been conducted using the UK Biobank resource under application number 49823. We are most grateful to the Bioinformatics Core Facility of Nantes BiRD, member of Biogenouest, Institut Français de Bioinformatique (IFB) (ANR-11-INBS-0013) for the use of its resources and for its technical support.

## Author contributions

**Conceptualization:** Thea Bismo Strøm, Katrine Bjune.

**Data curation:** Erik Kristoffer Arnesen, Antoine Rimbert, Anne Elise Eggen, Katrine Bjune.

**Formal analysis:** Åsa Schawlann Ølnes, Marianne Teigen, Katrine Bjune.

**Funding acquisition:** Katrine Bjune.

**Investigation:** Åsa Schawlann Ølnes, Marianne Teigen, Thea Bismo Strøm.

**Methodology:** Åsa Schawlann Ølnes, Marianne Teigen, Thea Bismo Strøm.

**Project administration:** Thea Bismo Strøm.

**Supervision:** Thea Bismo Strøm, Katrine Bjune.

**Visualization:** Åsa Schawlann Ølnes, Marianne Teigen, Katrine Bjune.

**Writing – original draft:** Katrine Bjune.

**Writing – review & editing:** Åsa Schawlann Ølnes, Marianne Teigen, Thea Bismo Strøm, Erik Kristoffer Arnesen, Antoine Rimbert.

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
