## [Decision Letter · Decision Letter 0]

13 Jan 2026

Dear Dr. Ølnes,

Thank you for submitting your manuscript to PLOS ONE. After careful consideration, we feel that it has merit but does not fully meet PLOS ONE’s publication criteria as it currently stands. Therefore, we invite you to submit a revised version of the manuscript that addresses the points raised during the review process.

We look forward to receiving your revised manuscript.

Kind regards,

Chiara Pavanello

Academic Editor

PLOS One

Journal Requirements:

“We would like to express our gratitude to the Norwegian Health Association for their financial support 484 of this study. We also thank the Tromsø Study for granting access to data and DNA samples from the 485 210 individuals with extreme HDL-C levels. Our appreciation extends to Kjetil Retterstøl and the Lipid 486 Clinic in Norway, as well as to Fürst Medical Laboratory and its director, Håvard Selby Ebbestad, for 487 providing the HDL-C distribution data. The present research has been conducted using the UK Biobank 488 resource under application number 49823. We are most grateful to the Bioinformatics Core Facility of 489 Nantes BiRD, member of Biogenouest, Institut Français de Bioinformatique (IFB) (ANR-11-INBS490 0013) for the use of its resources and for its technical support.”

“This research was supported by the Norwegian Health Association (https://nasjonalforeningen.no/) (Grant No. 22741 received by K.B.). The funding body had no role in study design, data collection and analysis, decision to publish, or preparation of the manuscript.”

4. We note that there is identifying data in the Supporting Information file  “S1_Table.pdf” Due to the inclusion of these potentially identifying data, we have removed this file from your file inventory. Prior to sharing human research participant data, authors should consult with an ethics committee to ensure data are shared in accordance with participant consent and all applicable local laws.

-Location data

Reviewer's Responses to Questions

**Comments to the Author**

1. Is the manuscript technically sound, and do the data support the conclusions?

Reviewer #1: Yes

Reviewer #2: Yes

2. Has the statistical analysis been performed appropriately and rigorously?

Reviewer #1: Yes

Reviewer #2: Yes

3. Have the authors made all data underlying the findings in their manuscript fully available?

Reviewer #1: Yes

Reviewer #2: Yes

4. Is the manuscript presented in an intelligible fashion and written in standard English?

Reviewer #1: Yes

Reviewer #2: Yes

Reviewer #1: Ølnes et al. investigated the genetic basis of extreme HDL-C levels in a population-based cohort in Northern Norway, using data from the Tromsø Study. This is a well-conducted and clearly presented study that investigates an underexplored area of lipid genetic: HDL-related dyslipidemia. The integration of functional assays and UK Biobank data strengthens the interpretation of the identified variants. The statistical analyses conducted are appropriate for the study design and objectives. Overall, the manuscript is well written and structured.

Minor comments:

- In some cases, the term “HDL” is used instead of “HDL-C” to describe HDL concentration. For clarity, consider using “HDL-C” when referring to measured concentrations

- On page 4, the authors state “In Norway, patients with extreme HDL-C levels have been offered genetic testing of the five key genes”, whereas in this study six genes are analyzed. Please clarify the reason for this difference

- On page 4, the authors report that approximately 270 individuals with extreme HDL-C levels and more than 70,000 individuals with abnormal LDL-C levels have been genetically analyzed in Norway over the past 25 years. Please clarify if the reported numbers are based on previously published data or reflect the authors’ own clinical experience

- Page 19, line 373: for consistency with the rest of the manuscript, consider replacing the term “mutations” with “variants”, and formatting the gene symbols according to standard gene nomenclature

Reviewer #2: This study aims to assess the prevalence of genetic variants responsible for extremely low and high levels of HDL-C using longitudinal data from the Tromsø Study, a population-based cohort in Northern Norway. In 210 individuals with extreme HDL-C levels, 38 variants of interest across six HDL-related genes were identified, of which 10 were considered potentially causative, found in 14 individuals. Sex-specific analyses showed that using HDL-C thresholds aligned with population distributions improved detection of individuals with pathogenic variants. These findings are clinically important in the context of raising awareness of the deleterious associations of extreme HDL-C with cardiovascular disease. The data are sound and the manuscript is well-written. However, the authors still need to address a short list of concerns as listed below.

Major points

1. Methods: The definitions of extremely low and high HDL-C levels appear arbitrary and should be supported by relevant references.

2. Results: The description of the supplementary data (Figures S1 to S6) in the text of the manuscript is unsatisfactory. These data should be presented in more details. Figure S6 should be described in the Results section, not in the Discussion.

Minor points

1. Introduction and throughout the manuscript: The terms “HDL levels” and HDL-C levels” seem to be used as synonyms, which is not always appropriate (e.g. lines 76-77). Please revise.

2. Results: Table S4 contains important data on the pathogenicity of genetic variants and should be moved to the main manuscript.

3. Discussion, p. 21: LPL directly contributes to the circulating HDL pool by producing surface remnants of triglyceride-rich lipoproteins rich in phospholipids and free cholesterol. Please revise.

4. The Discussion section is too lengthy and should be shortened.

.

Reviewer #1: No

Reviewer #2: No

---

## [Author Response · Author response to Decision Letter 1]

4 Feb 2026

Response to comments from the reviewers on Manuscript Number: PONE-D-25-62448

Response to Reviewer #1

We thank Reviewer #1 for the thorough and positive evaluation of our manuscript, and for the constructive comments. We are pleased that the reviewer finds the study well conducted, clearly presented, and strengthened by the integration of functional assays and UK Biobank data. Below, we address each comment in detail.

Minor comments

1. “In some cases, the term ‘HDL’ is used instead of ‘HDL-C’ to describe HDL concentration. For clarity, consider using ‘HDL-C’ when referring to measured concentrations.”

Response:

We agree and thank the reviewer for pointing this out. The manuscript has been carefully revised to ensure consistent use of “HDL-C” whenever referring to measured cholesterol concentrations, while “HDL” is reserved for particle/function-related descriptions. These changes have been implemented throughout the text.

2. “On page 4, the authors state that patients with extreme HDL-C levels have been offered genetic testing of five key genes, whereas six genes are analyzed in this study. Please clarify the reason for this difference.”

Response:

Thank you for highlighting this important point. We have now clarified this explicitly in the Introduction.

Briefly, PLTP is not part of the routine diagnostic gene panel for HDL-related dyslipidemia at our unit. However, in the context of this study, we had sufficient DNA material to sequence one additional gene beyond the standard panel. We therefore chose to include PLTP for two main reasons:

1. We anticipated that PLTP variants could be assessed using the same experimental system as CETP variants (although this later proved not to be feasible for functional assays), and

2. PLTP is relatively understudied in the context of extreme HDL-C phenotypes, and we aimed to explore its potential contribution and variant frequency in a population-based setting.

This rationale has now been clearly stated in the revised manuscript (Introduction), to avoid confusion between routine diagnostics and the research-based gene selection in this study.

3. “Please clarify if the reported numbers are based on previously published data or reflect the authors’ own clinical experience.”

Response:

We have now added an explicit statement in the revised manuscript clarifying that these numbers do not represent previously published data, but instead reflect diagnostic activity at our clinical genetic unit.

Specifically, the figures refer to patients referred for and analyzed by genetic testing at the Unit for Cardiac and Cardiovascular Genetics, Oslo University Hospital, over the past 25 years. This additional clarification has been included in the Introduction to ensure transparency regarding the data source.

4. “For consistency, consider replacing the term ‘mutations’ with ‘variants’, and formatting gene symbols according to standard gene nomenclature.”

Response:

We agree with this comment. The manuscript has been revised accordingly:

The term “mutations” has been replaced with “variants” throughout the text, and all gene symbols are now consistently formatted according to standard gene nomenclature.

Once again, we thank Reviewer #1 for the careful reading of our manuscript and for the constructive suggestions, which have helped improve clarity and precision.

Response to Reviewer #2

We thank Reviewer #2 for the careful reading of our manuscript and for the positive assessment of the data quality, clinical relevance, and clarity of the presentation. We particularly appreciate the recognition of the clinical importance of raising awareness of the deleterious associations of extreme HDL-C levels with cardiovascular disease. Below, we address each of the reviewer’s concerns in detail.

Major points

1. “The definitions of extremely low and high HDL-C levels appear arbitrary and should be supported by relevant references.”

Response:

We thank the reviewer for raising this point. We agree that, without adequate explanation, the HDL-C thresholds used in this study (≤ 0.5 mmol/L and ≥ 3.0 mmol/L) may indeed appear arbitrary, and we acknowledge that this was not sufficiently clear in the original version of the manuscript.

These thresholds were not derived from population-based percentiles, but were historically defined by clinicians at our diagnostic unit more than a decade ago, within a markedly different clinical and scientific landscape. At that time, genetic testing was resource-intensive and costly, laboratory capacity was limited, and clinical awareness of HDL-related disorders was low. Healthcare systems, including the Norwegian public healthcare system, emphasized cost–benefit considerations, avoidance of unnecessary testing, and prioritization of limited genetic resources toward conditions with established therapeutic consequences. Together, these factors favored the use of particularly strict eligibility criteria for genetic testing.

Importantly, these stringent thresholds were selected in clinical settings to enrich for individuals with a high likelihood of monogenic causes of extreme HDL-C levels. We have now added references supporting the biological relevance of the applied cutoffs and reflecting the evidence base that informed the original diagnostic criteria.

Since then, the clinical and scientific landscape has changed substantially. There is now increased recognition of the association between extreme HDL-C levels and adverse outcomes, greater interest in HDL biology, and markedly reduced costs and increased capacity for genetic testing.

We fully agree with the reviewer that these historical thresholds were unusually strict. Importantly, recognizing this limitation is a central motivation for the present study. By leveraging population-based data, we demonstrate that the previous cutoffs introduced a pronounced sex bias and captured only a small fraction of individuals with potentially pathogenic variants. In response, and partly informed by the work conducted during the preparation of this manuscript, the criteria for clinical genetic testing at our unit have now been revised toward less extreme, sex-specific thresholds. The Methods and Discussion sections have been revised to more clearly explain the origin of the original cutoffs, their limitations, the clinical reasoning underlying the updated thresholds, and the references used.

2. “The description of the supplementary data (Figures S1 to S6) in the text of the manuscript is unsatisfactory. These data should be presented in more detail. Figure S6 should be described in the Results section, not in the Discussion.”

Response:

The supplementary figures are now described in greater detail in the Results section, with explicit references to the relevant figures. And to improve logical flow and clarity, the supplementary figures have been renumbered: the former S6 Fig is now S1 Fig, and the previous S1–S5 Figs are now S2–S6 Figs.

Minor points

1. “The terms ‘HDL levels’ and ‘HDL-C levels’ seem to be used as synonyms.”

Response:

We agree with this comment. The manuscript has been revised to ensure consistent and appropriate use of terminology: “HDL-C” is now used when referring to measured cholesterol concentrations, whereas “HDL” is reserved for particle- or function-related contexts.

2. “Table S4 contains important data on the pathogenicity of genetic variants and should be moved to the main manuscript.”

Response:

We appreciate the reviewer’s suggestion and understand the rationale for highlighting these data. Table S4 summarizes genetic variants previously identified at our diagnostic unit and is intended to provide additional clinical context based on our cumulative experience with HDL-related disorders. These variants are independent of the Tromsø Study cohort and do not originate from the population-based material analyzed in the present study.

For this reason, we chose to place Table S4 in the supplementary material, in order to clearly distinguish historical diagnostic findings from the primary results derived from the Tromsø Study. We felt that including these data in the main manuscript might risk blurring this distinction.

That said, we acknowledge the reviewer’s point that the information may be of interest to readers. If the reviewer or editor considers it important to move Table S4 to the main manuscript, we would of course be happy to do so or to revise its presentation accordingly.

3. “LPL directly contributes to the circulating HDL pool by producing surface remnants of triglyceride-rich lipoproteins rich in phospholipids and free cholesterol. Please revise.”

Response:

Thank you for this clarification. When shortening the Discussion section (Minor comment #4), we decided to remove the description of the roles of each protein mentioned in “Limitations of the study” (ANGPTL3, LPL, LAL, hepatic and endothelial lipase). If the reviewer or editor finds it important to include these descriptions, we will include them in the manuscript and make sure that LPL’s role is correctly explained.

4. “The Discussion section is too lengthy and should be shortened.”

Response:

We agree and have substantially revised the Discussion section. The total length has been reduced from 2264 words to 1823 words, corresponding to an approximate 20% reduction, while preserving the key clinical, methodological, and interpretative points. Redundant passages have been removed or condensed, and the overall structure has been tightened to improve readability.

Once again, we thank Reviewer #2 for the thoughtful and constructive feedback, which has led to improvements in clarity, structure, and contextualization of the manuscript.

Response to Academic Editor

We thank the Academic Editor for the careful assessment of our manuscript and for the clear and constructive guidance, which has helped us improve the manuscript’s compliance with PLOS ONE’s policies and standards.

Response:

We have carefully reviewed the manuscript and all associated files and ensured that they comply with PLOS ONE’s style and file-naming requirements.

Response:

All funding-related text has been removed from the manuscript. The amended Funding Statement has been included in the revised cover letter, as requested.

“This research was supported by the Norwegian Health Association (https://nasjonalforeningen.no/) (Grant No. 22741 received by K.B.). The funding body had no role in study design, data collection and analysis, decision to publish, or preparation of the manuscript.”

Response:

The amended statement has been included in the revised cover letter.

4. We note that there is identifying data in the Supporting Information file “S1_Table.pdf” Due to the inclusion of these potentially identifying data, we have removed this file from your file inventory. Prior to sharing human research participant data, authors should consult with an ethics committee to ensure data are shared in accordance with participant consent and all applicable local laws.

Data sharing should never compromise participant privacy. It is therefore not appropriate to publicly share personally identifiable data on human research participants.

Response:

We have revised S1 Table to further anonymize the data by replacing specific participant ages with age groups and have ensured that the data shared are in accordance with participant consent and applicable regulations. The updated, anonymized S1 Table has been re-uploaded to the file inventory. Corresponding changes have also been made in Table 2 in the main manuscript.

Response:

We have removed the phrase “data not shown” and the corresponding statement from the figure legend of S3 Fig, as the information was not essential to the presentation or interpretation of the results.

Response:

The ethics statement has been revised in the Methods section to explicitly state that written informed consent was obtained from all participants.

Response:

No specific recommendations to cite additional publications were made by the reviewers.

Response:

The reference list has been carefully reviewed. No retracted articles are cited.

Additional changes:

In addition to the revisions described above, we have carefully reviewed and refined the pathogenicity interpretation of several genetic variants based on constructive feedback received from a member of a PhD defense committee, who had access to the manuscript through its inclusion in a doctoral dissertation. Corresponding updates have been made to Tables S2–S4. These refinements enhance the scientific rigor and clarity of the manuscript and do not alter the main findings or conclusions of the study.

---

## [Decision Letter · Decision Letter 1]

23 Feb 2026

Genetic testing in individuals with extreme HDL-C levels: diagnostic yield and clinical implications from the Tromsø Study

PONE-D-25-62448R1

Dear Dr. Ølnes,

We’re pleased to inform you that your manuscript has been judged scientifically suitable for publication and will be formally accepted for publication once it meets all outstanding technical requirements.

Kind regards,

Chiara Pavanello

Academic Editor

PLOS One

Additional Editor Comments (optional):

Reviewers' comments:

Reviewer's Responses to Questions

**Comments to the Author**

Reviewer #1: All comments have been addressed

Reviewer #2: All comments have been addressed

2. Is the manuscript technically sound, and do the data support the conclusions?

Reviewer #1: (No Response)

Reviewer #2: Yes

3. Has the statistical analysis been performed appropriately and rigorously?

Reviewer #1: (No Response)

Reviewer #2: (No Response)

4. Have the authors made all data underlying the findings in their manuscript fully available?

Reviewer #1: (No Response)

Reviewer #2: Yes

5. Is the manuscript presented in an intelligible fashion and written in standard English?

Reviewer #1: (No Response)

Reviewer #2: Yes

Reviewer #1: (No Response)

Reviewer #2: All comments have been addressed. I have no further suggestions and thank the authors for their work.

.

Reviewer #1: No

Reviewer #2: No

---

## [Editor Report · Acceptance letter]

PONE-D-25-62448R1

PLOS One

Dear Dr. Ølnes,

I'm pleased to inform you that your manuscript has been deemed suitable for publication in PLOS One. Congratulations! Your manuscript is now being handed over to our production team.

Kind regards,

on behalf of

Dr. Chiara Pavanello

Academic Editor

PLOS One